# The Association between Birth Satisfaction and the Risk of Postpartum Depression

**DOI:** 10.3390/ijerph181910458

**Published:** 2021-10-05

**Authors:** Eva Urbanová, Zuzana Škodová, Martina Bašková

**Affiliations:** Department of Midwifery, Jessenius Faculty of Medicine, Comenius University, 036 01 Martin, Slovakia; eva.urbanova@uniba.sk (E.U.); zuzana.skodova@uniba.sk (Z.Š.)

**Keywords:** risk of postpartum depression, birth satisfaction, stress, postpartum period

## Abstract

Negative experiences with childbirth might have a negative impact on a woman’s overall health, including a higher risk of postpartum depression. The aim of the study was to examine the association between birth satisfaction and the risk of postpartum depression (PPD). A 30-item version of the Birth Satisfaction Scale (BSS) and the Edinburgh Postnatal Depression Scale (EPDS) were used, as well as the Perceived Stress Scale (PSS). The study included 584 women (mean age 30.6 ± 4.9), 2 to 4 days postpartum. In the regression model, the negative effect of birth satisfaction on the risk of postpartum depression was shown: a lower level of satisfaction with childbirth was a significant predictor of a higher risk of PPD (β = −0.18, 95% CI = −0.08; −0.03). The regression model was controlled for the effect of the sociodemographic factors (such as education or marital status) and clinical variables (such as parity, type of delivery, psychiatric history, levels of prenatal stress). Levels of prenatal stress (β = 0.43, 95% CI = 0.27; 0.39), psychiatric history (β = 0.08, 95% CI = 0.01; 3.09), parity (β = −0.12, 95% CI = −1.82; −0.32) and type of delivery (β = 0.11, 95% CI = 0.20; 1.94) were also significantly associated with the levels of postnatal depression. The current study confirmed the association between the level of birth satisfaction and the risk of developing PPD, i.e., a lower satisfaction with childbirth may increase the risk of developing PPD.

## 1. Introduction

Hollins, Martin, and Fleming identified three basic areas related to childbirth satisfaction, which they transformed into an effective measurement tool: service provision (home assessment, birth environment, support, relationships with health care professionals); personal attributes (ability to cope during labor, feeling in control, childbirth preparation, relationship with baby); and stress experienced during labor (distress, obstetric injuries, receiving sufficient medical care, obstetric intervention, pain, long labor and baby’s health) [1]. Satisfaction with childbirth is multifactorial. In addition to the above-mentioned factors, women also evaluate the events that happened on the particular day of childbirth [2,3].

The overall experience of maternity care can also have a positive or negative impact on the health and emotional well-being of a woman, a child and the wider family system, whereby the impact can be immediate or long term [4,5]. A positive birth experience can support a woman’s personal growth, self-esteem and successful motherhood [6,7]. A negative birth experience increases the risk of negative health outcomes, such as postpartum depression and future fear of giving birth, which can lead to a request for caesarean birth in future pregnancies, and have an impact on future reproduction in the sense of refusing another pregnancy [7,8].

Positive childbirth experiences can act as an effective prevention of psychological postpartum trauma. In a systematic review and meta-analysis of prenatal and intrapartum interventions, Taheri et al. define effective strategies that shape a positive perception of childbirth: support for women, minimal interventions in the birth process and birth preparedness. However, interestingly, strategies for relaxation and pain relief do not fall into this category [9]. To a large extent, Funai et al. list the four most important factors determining mothers’ satisfaction with childbirth, such as her personal expectations, support that she receives during childbirth, the quality of the caregiver–patient relationship (for instance, respect, communication, continuity of care) and participation in decision making [10]. High-quality and individualized maternity care during childbirth is a factor associated with very positive birth satisfaction in the Swedish study by Hildingsson, Johansson, Karlström and Fenwick [11]. The WHO (World Health Organization) recognizes that promoting a positive childbirth experience is a very important factor in maternal care, and introduces a global list of recommendations for Intrapartum Care for a Positive Childbirth Experience. Some of these recommendations are directly related to lower PPD (postpartum depression) incidence, such as the selection of the support person who is not a member of hospital staff [12].

Approximately 2–4% of women after childbirth develop post-traumatic stress disorder (PTSD), which is also strongly associated with the level of childbirth satisfaction and which has potentially far-reaching negative consequences for the quality of the couple’s relationship [13,14,15]. Maternal PTSD further leads to low self-esteem, lower birth weight and difficulty breastfeeding as well as problems with a woman’s sexuality [16]. In many cases, the occurrence of PTSD symptoms also predicts the occurrence of postpartum depressive symptoms [14].

Women’s satisfaction with childbirth, as a very important attribute of obstetric care, can be objectively measured. Patient satisfaction is an important result of the evaluation and development of health services [17]. Thus, measuring women’s satisfaction with their births can provide effective feedback on correcting gaps in care provided [18] or reveal contexts where birth satisfaction is a significant risk factor.

Many research studies have confirmed that birth satisfaction can be a significant predictor of the increased risk of postpartum depression (PPD) [7,19,20,21].

Postpartum depression is a disease with clearly defined diagnostic criteria listed in the International Classification of Diseases, where the main symptoms include mood swings, decreased interest in daily activities, fatigue and loss of energy, difficulty concentrating, serious worries about the child, sadness and crying, expressing doubts, decreased libido, appetite disorders, sleep disorders, or even thoughts of death and suicide [22,23]. The global prevalence of PPD is approximately 17%, with significant heterogeneity across nations [24]. It is important to identify the risk of postpartum depression in a timely manner using effective screening tools, as PPD is associated with immediate and persistent adverse effects on maternal and offspring mortality and morbidity [25].

In addition to birth satisfaction, stress is also a significant factor contributing to PPD. A study by Scheyer and Urizar showed that increased stress levels, especially in the early stages of pregnancy, subsequently lead to a risk of developing PPD. Factors affecting stress probably are: low-income in pregnant women and related childbirth responsibilities, fear of parenthood and lack of social support [26]. However, the role of stress in the development of PPD is even more significant, as suggested by the systematic review by Yim, Tanner, Stapleton, and Guardino et al. [27]. Additionally, aspects directly related to pregnancy and childbirth, such as parity, type of childbirth, positive psychiatric history and others, may be involved in the development of PPD [28,29,30,31].

The aim of our study is to determine the association between childbirth satisfaction and the risk of postpartum depression, controlling for the effect of potential factors that could increase the risk of postpartum depression: sociodemographic factors (such as education or financial situation), and clinical factors (such as levels of prenatal stress, type of birth, parity or psychiatric history). Other objectives of the study were the relationships in the subgroups between stress, type of childbirth, parity, positive psychiatric history and the risk of postpartum depression as well as the relationships in the above-mentioned subgroups and birth satisfaction.

In particular, we would like to point out the importance of the study from the point of view of the country where it was carried out. This topic is explored minimally or not at all in post-communist countries. Post-communist countries have specific problems in providing quality maternity care. It is very important to highlight this issue.

## 2. Materials and Methods

### 2.1. Research Design

A cross-sectional correlational design was used.

### 2.2. Setting and Participants

The research was carried out in two university hospital birth centers, which are located in the capital (Bratislava) and in the central part of Slovakia (Martin). A convenience sampling method was chosen to select participants. In addition to the postpartum period, the inclusion criteria were the birth of a living child, willingness to cooperate and informed consent. The research sample consisted of 584 women (mean age 30.6 ± 4.9, age range 16–44) at two to four days after giving birth, i.e., in the early postpartum period, when a rapid decrease in hormones affects CNS (Central Nervous System) function [25]. In total, 381 women (65.2%) came from Martin University Hospital and 203 women (34.8%) participated in Bratislava University Hospital.

The management of each university hospital acknowledged the approval of the Ethics Committee of Jessenius Faculty of Medicine in Martin, Slovakia, no. EK 36/2018. Based on this fact, the heads of the Gynecology and Obstetrics Departments agreed to conduct the research and did not require any further permission from the hospital ethics committees. More detailed characteristics of the research sample are given in Table 1.

### 2.3. Data Collection

Data collection took place between September 2018 and February 2020 via paper-and-pencil questionnaires administered by midwives in a clinical setting. Participants received information about study aims, and each participant signed the informed consent form before administering the questionnaire. After obtaining consent, the midwife completed the date of collection and the specific day after giving birth in each participant’s questionnaire. The response rate of the questionnaires was 82.3%.

### 2.4. Measuring Instruments

The original 30-item version of the Birth Satisfaction Scale (BSS) by C.J. Hollins Martin was used to assess birth satisfaction. The BSS contains 30 self-report items, rated on a 5-point Likert scale, that assess women’s perceptions of: (1) quality of care provision; (2) women’s personal attributes; and (3) stress experienced during labor—8, 8 and 14 items per factor [1]. The total possible range of scores is 30 to 150 points. A score of 30 points represents the least satisfied and a score of 150 points represents the most satisfied with childbirth. In the present study, the internal consistency of the BSS was 0.89 (Cronbach’s α = 0.89), which indicates good reliability. The internal consistency of each subscale is shown in Table 2. Permission to use the BSS has been obtained.

The risk of postpartum depression in respondents was assessed using the Edinburgh Postnatal Depression Scale (EPDS). The scale consists of 10 short statements. A mother checks off one of four possible answers that is closest to how she has felt during the past week. Responses are scored 0, 1, 2 and 3 based on the seriousness of the symptom. The total score is found by adding together the scores for each of the 10 items [32]. In this study, different cut-off points have been used. According to the EPDS manual, 2nd edition (2014), a cut-off score of 10 points or more is recommended for research, indicating elevated levels of depressive symptoms. In clinical use, a cut-off score of 13 has been shown to detect major depression, and a woman who meets this threshold should be further assessed by a mental health professional, because the Edinburgh Postnatal Depression Scale is used as a routine screening instrument [33]. In the present study, the internal consistency of the EPDS was 0.84 (Cronbach’s α = 0.84), which indicates good reliability. Permission to use the EPDS has been obtained from the Royal College of Psychiatrists (UK).

Another research tool used in our study was the 10-item Perceived Stress Scale (PSS-10), which is the most widely used psychological instrument for measuring the perception of stress. It is a measure of the degree to which situations in human life are assessed as stressful. The PSS was designed as a measurement tool for use in different communities, as the items are generally designed and easy to understand. The questions in this scale ask about feelings and thoughts during the last month. In each case, respondents are asked how often they felt a certain way [34]. For the purposes of this study, the questions in the PSS examine women’s feelings during pregnancy. Each item on the PSS-10 is rated on a 5-point Likert scale, ranging from 0 (never) to 4 (very often). The PSS-10 consisted of six positively (items 1, 2, 3, 6, 9 and 10: Positive factor) and four negatively worded items (items 4, 5, 7 and 8: Negative factor). Total scores range from 0 to 40, with higher scores indicating higher levels of perceived stress. Scores ranging from 0 to 13 would be considered as low stress. Scores ranging from 14 to 26 would be considered as moderate stress and scores ranging from 27 to 40 would be considered as high perceived stress [34,35]. In the present study, the internal consistency of the PSS was 0.83 (Cronbach’s α 0.83), which indicates good reliability.

### 2.5. Data Analysis

Statistical data analysis was performed using PSPP (GNU Project, version 1.4.1) and IBM SPSS Statistics for Windows (IBM, version 27.0, Endicott, NY, USA). Descriptive statistics were used to describe the basic characteristics. A hierarchical linear regression model (enter method) was used to determine whether birth satisfaction predicts levels of postpartum depression, while controlling for sociodemographic factors (age, education, financial situation, religiosity, marital status) and clinical factors (prenatal stress, psychiatric history, chronic health condition during pregnancy, newborn disease, support person at birth, preterm birth, parity, type of birth, gestational diabetes, perinatal loss in the previous anamnesis). Student’s t-test for independent samples was used to determine differences in the mean score of the risk of depression and childbirth satisfaction between key variables. The statistical processing also included determination of the internal consistency of the scales used (research tools) by means of the Cronbach’s alpha coefficient.

## 3. Results

### 3.1. Overall Assessment of Birth Satisfaction, the Risk of Postnatal Depression and the Level of Stress

In the 30-item BSS, the respondents achieved a total mean score of 111.14 ± 15.45 (min. 61.00–max. 149.00) out of a maximum score of 150, i.e., the highest possible childbirth satisfaction. The highest satisfaction was achieved in the subscale quality of care in the percentage conversion of the achieved score to the maximum score (77.05%), followed by the subscale women’s personal attributes (73.6%) and the lowest satisfaction was recorded in the subscale stress experienced during labor (72.61%).

In the whole sample, the risk of postnatal depression, which was investigated using the EPDS tool, achieved an average score of 5.24 ± 4.35. Ninety-six women scored 10 points or more. In 35 women, the cut-off score was more than 13 points, i.e., the clinical criterion where there is a high risk of major depressive disorder.

In the whole sample, the level of stress experienced during pregnancy and childbirth was investigated using the Perceived Stress Scale with a mean score of 14.18 ± 5.68, which is considered moderate stress. A more detailed evaluation of specific scales is given in Table 3.

### 3.2. The Effect of Birth Satisfaction on the Risk of Postpartum Depression

A linear regression model, which allows for multivariate analysis, i.e., evaluation of the association between several factors and the outcome variable (in our case, the risk of postpartum depression) was used to examine the effect of birth satisfaction on the risk of postpartum depression. In the regression model, the main predictor (the risk factor) was birth satisfaction, and the model was controlled for the effect of potential confounding variables: sociodemographic factors (age, education, financial situation, religiosity, marital status), and clinical factors (prenatal stress, psychiatric history, chronic health condition during pregnancy, newborn disease, support person at birth, preterm birth, parity, type of birth, gestational diabetes, perinatal loss in the previous anamnesis).

For the purposes of statistical analysis, the dichotomous variables inserted into the model were coded as follows: psychiatric history—yes/no; chronic disease in pregnancy—yes/no; newborn disease—yes/no; support person—yes/no; preterm labor—yes/no; parity—primipara/multipara; type of birth—vaginal/operative; gestational diabetes—yes/no; perinatal loss—yes/no; education—primary/secondary/higher; financial status—very good/quite good/not very good; religiosity participation (for religiosity, we asked how often a woman actively participates in religious activities)—no/sometimes/regularly; marital status—with a partner/without a partner.

In regression model 1, the association between sociodemographic factors and the risk of postpartum depression was investigated (Model 1). Age (β = −0.10, 95% CI = −0.18; −0.01) and financial status (β = 0.14, 95% CI = 0.37; 1.80) were significantly associated with the postpartum depression levels in this model. The adjusted R2 of this model was 0.03.

Clinical variables were added in model 2, where the adjusted R2 value was 0.30, and the effects of prenatal stress (β = 0,48, 95% CI = 0.30; 0.43), psychiatric history (β = 0.09, 95% CI = 0.23; 3.34), newborn disease (β = 0.09, 95% CI = 0.56; 7.14), parity (β = −0.13, 95% CI = −1.98; −0.47) and type of birth (β = 0,18, 95% CI = 1.06; 2.64) were statistically significant.

In the final regression model (Model 3), the effect of both sociodemographic and clinical factors together with birth satisfaction was assessed. In this model, the associations between EPDS levels and the following variables were statistically significant: birth satisfaction (β = −0.18, 95% CI = −0.08; −0.03), stress (β = 0.43, 95% CI = 0.27; 0.39), psychiatric history (β = 0.08, 95% CI = 0.01; 3.09), parity (β = −0.12, 95% CI = −1.82; −0.32) and type of delivery (β = 0.11, 95% CI = 0.20; 1.94). The total explained variance (adjusted R2 value) in this model was 0.32 (Table 4).

### 3.3. Differences in the Incidence of Postpartum Depressive Symptoms Based on Perinatal History Factors

Student’s t-test showed that respondents with a positive psychiatric history had a statistically significant risk of postpartum depression (*p* ≤ 0.000) and, at the same time, statistically significant lower birth satisfaction (*p* ≤ 0.003) than respondents with a negative psychiatric history.

Furthermore, concerning the type of delivery (vaginal/natural vs. operative delivery), there was a statistically significant level of maternal postpartum depression after caesarean delivery (*p* ≤ 0.000) than after natural birth. At the same time, there was a statistically significant lower birth satisfaction after caesarean delivery (*p* ≤ 0.000) than after normal birth.

In terms of parity, primiparous women had a statistically significant higher level of postpartum depression (*p* ≤ 0.020) and lower birth satisfaction (*p* ≤ 0.017) than multiparous ones. The average score between EPDS and BSS in these groups is shown in Table 1.

## 4. Discussion

The aim of the present study was to explore the relationship between childbirth satisfaction and the level of postpartum depression, considering other factors that could affect the risk of postpartum depression.

### 4.1. Birth Satisfaction

Childbirth satisfaction, as the main predictor of the risk of postpartum depression, reached an average value of 111.14 in the monitored group. In the Slovak Republic, this score can be evaluated as stable, as in the study from 2019 [36], which focused on validating the Slovak version of the 30-item BSS scale, the average value of birth satisfaction was 112. Thus, the results are almost identical with those of the current study, with a very similar number of respondents (there were more than 500 respondents in both studies). In isolation, it is difficult to assess whether the level of satisfaction achieved is an acceptable value. Only slightly higher birth satisfaction scores (115.5) were reported in a Canadian study on maternal and paternal satisfaction, whereas the score in the mothers was evaluated as relatively high [37].

The results show that birth satisfaction in the group varied depending on several factors, which included the level of stress experienced during pregnancy, the type of childbirth, parity and a positive psychiatric history.

Statistical analysis in our study confirmed that the higher level of stress experienced during pregnancy and childbirth was significantly associated with birth satisfaction. Hinic presents similar findings, but in a smaller sample size [38]. Major life changes, such as childbirth, are stressful. The level of stress experienced during pregnancy is influenced by various factors, including dissatisfaction with the relationship with the partner or childbirth-related fear [39,40,41] and ultimately, they are reflected in postpartum outcomes, which include birth satisfaction.

Significantly lower satisfaction was also shown in our group of respondents with surgical delivery in comparison with respondents with physiological, vaginal delivery. This outcome is confirmed by several studies [7,37,42,43,44,45,46,47,48], which present medical interventions, cesarean sections and various other surgical interventions during childbirth as predictors of lower birth satisfaction. Prior to any instrumental intervention in the birthing process, a proper assessment of the situation should be made on the basis of credible evidence so that the benefits of the procedure outweigh the negative side effects [42].

Parity in our sample was recoded into primiparas (60%) and multiparas (40%), with primiparas showing significantly lower birth satisfaction than multiparas. In a cohort study including 16,000 postpartum women with a focus on obstetric interventions by Falk, Nelson and Blomberg, lower birth satisfaction in primiparous women was assessed/proven [40]. On the contrary, higher birth satisfaction in multiparous women is presented in a study based on WHO standards by Lazzerini, Mariani, Semenzato et al. [49].

Parity has a significant effect on the overall birthing process. In primiparous women, a labor progression is slower than in multiparous [50]. Thus, the length of labor may be a decisive factor for women’s childbirth satisfaction [51]. As part of the reinforcement of a positive childbirth experience, according to the WHO, women should be informed that the duration of the first stage of labor can vary from one woman to another, but is usually longer in first-time mothers [12].

Some studies also confirmed an association between a positive psychiatric history or various mental health problems during pregnancy and birth satisfaction [49,52]. Our study sample also confirmed that women with a positive psychiatric history have a lower level of childbirth satisfaction (*p* ≤ 0.003) than respondents with a negative psychiatric history. Pregnancy is a period of increased vulnerability for the development of anxiety and depressive episodes [53], which can lead to reduced mental resilience and ineffective coping with labor.

### 4.2. Risk of Postpartum Depression Symptoms

The prevalence of the onset of depressive symptoms with the EPDS at ≥10 points (research criterion) was 16.7% and the prevalence of severe depressive symptoms (EPDS ≥13 points) was 6.07%. Previous studies conducted in Slovakia found a prevalence of PPD (major depressive symptoms) in the early postpartum period ranging from 14 up to 25.3% using various research tools [54,55]. Based on data from the Czech version of the European Longitudinal Study of Pregnancy and Childhood, the prevalence of PPD in the 6-week postpartum period (EPDS ≥10 points) was 21.9%, but with a stricter EPDS score, prevalence was 11.8% [31]. The results of a large and prospective multicenter cohort study of 3310 women from France, where the prevalence of PPD by 8 weeks postpartum was 8.3% [56], are the closest to our current findings. According to Hahn-Holbrook, Cornwell-Hinrichs and Anaya, the global prevalence of PPD was 17% and varied dramatically by nation [24].

Multivariate analysis, used in this study, allowed associations to be examined between several factors related to postpartum depression. In addition to the strongest predictor, satisfaction with childbirth, other significant predictors were: the level of stress experienced during pregnancy, the type of childbirth, parity, and a positive psychiatric history.

A number of studies report lower birth satisfaction as a significant risk factor for PPD [7,8,19,20,21,57]. This result is also consistent with our findings, where lower birth satisfaction has been shown to be the strongest predictor of a higher risk of postpartum depression. The linear regression model showed childbirth satisfaction as a statistically significant factor for the higher risk of PPD. There are also findings that do not confirm such a relationship. According to Bell et al., for example, negative women’s childbirth experience increases anxiety but decreases the symptoms of PPD [45].

Biaggi et al. identified adverse life events and increased levels of perceived stress as the main risk factors associated with PPD [58]. Similarly, Fiala et al. found that psychosocial stressors were also highly associated with developing PPD [31]. In a relatively large Canadian study, Chow et al. concluded that women with low scores on perceived stress were much less likely to suffer from depressive symptoms in the pre- and postnatal period [53]. In our sample, the regression model confirmed that the effect of stress as well as childbirth satisfaction significantly increases the risk of PPD and after adding stress variable to the regression model (Model 2), the total adjusted R2 value increased to 0.29.

The issue of stress and its impact is very broad. In a systematic review by Yim, Tanner Stapleton, Guardino, Hahn-Holbrook and Dunkel Schetter, focusing on the biological and psychological predictors of PPD, stressors are divided into two major groups: episodic stressors (stressful life events, catastrophic events, daily hassles) and chronic stressors (parenting stress, perceived stress, chronic strain often associated with a woman’s employment) [27]. We did not pay close attention to these aspects in our study, so we were not able to identify stressors more exactly. The higher level of perceived stress experienced during all three trimesters of pregnancy and also at 3 months postpartum was significantly associated with higher levels of PPD in low-income pregnant women [26], suggesting that early detection of perceived stress and stressors during pregnancy may later have a protective significance

In our sample, 74% of the respondents reported vaginal delivery and 26% had operative delivery. The type of birth seems to be significantly associated with PPD, too. There was a significantly higher risk of postpartum depression in the group of women after operative delivery than in the women who delivered vaginally. Our findings are similar to the other studies [8,20,59]. According to a meta-analysis by Xu, Ding, Ma, Xin and Zhang, caesarean section (CS) increases the risk of PPD, but further studies about the specific types of CS and PPD risk are needed [29]. According to Eckerdal et al., there is not any association between mode of delivery and PPD; however, if operative delivery is associated with women’s negative experience, the risk of PPD is higher [60]. Other studies have not confirmed this association with certainty [31,45,61].

The effect of parity on the postpartum period has not been sufficiently studied and the study results are inconsistent [28]. Several studies considered multiparity as a risk factor for PPD, especially in relation to higher childcare burden and psychosocial stress [31]. On the contrary, Martínez-Galiano et al. identified an increased risk of postpartum depression, anxiety and sadness symptoms in primiparous women [28]. Similar conflicting results have been shown in a systematic review of the risk of antenatal depression and anxiety; some studies have reported increased risk of developing antenatal anxiety and depression in multiparas, while other studies found primiparous women to be more at risk than multiparas [58]. Our results confirmed higher levels of postpartum depression in primiparous than in multiparous women, which is consistent with other studies linking primiparity to the risk of PPD.

According to the Centers for Disease Control and Prevention (CDC), previous episodes of depression, anxiety, and mood disorders are the primary risk factors for PPD [30]. In our study, we included episodes of depression before or during pregnancy and other mental illness in the positive psychiatric history. Student’s t-test confirmed that respondents with a positive psychiatric history had a statistically significant risk of postpartum depression than women with a negative history. There is strong evidence to support this claim. Fiala et al. confirmed 280% higher risk of developing PPD in women with a positive psychiatric history and even a significant correlation with a family history of depression on both sides (mother and father), which appeared in the group of women at 6 weeks postpartum [31]. Meyers et al. stated that past history of depression or anxiety, whether or not associated with a previous pregnancy, consistently increased the risk of postpartum depression (the strength of evidence was moderate) [62]. A large IGEDEPP (Interaction of Gene and Environment of Depression during PostPartum) prospective cohort study clearly confirmed that the personal history of any psychiatric disorder was significantly associated with the risk of early onset PPD (first 6 weeks postpartum) as well as late PPD (between the second and twelfth months postpartum), and a positive family psychiatric history was associated with a higher risk of developing late PPD [56].

Among other sociodemographic and clinical factors and the risk of PPD, the association between age, financial status and newborn disease was identified. No other significant association occurred between the other variables (e.g., education, marital status, preterm labor, chronic disease in pregnancy, perinatal loss, etc.), which were investigated using a linear regression model and the resulting variable, i.e., PPD.

### 4.3. Limitations of the Study

We consider a relatively small sample size to be a study limitation. Although the sample consisted of 584 women, several studies dealing with similar issues had a much higher sample size. As the research was conducted in two university hospitals in western and central Slovakia, women from the eastern part of the Slovak Republic are minimally represented in the sample. The lack of respondents with lower education can also be considered a very limiting factor. In our sample, they made up only 2.2% (*n* = 13). During data collection in the clinical setting, these women were reluctant to complete the questionnaire. Some of them even refused introductory verbal and written information about the research because of lack of interest in the research, shortage of time or care of the newborn. On the contrary, women with higher education, who made up more than 50% of the sample, showed the greatest interest in our research study. In the sample, education does not correspond to the educational structure of women in Slovakia, as most women have completed secondary education, followed by women with lower education and university degrees [63]. In several studies, education was recognized as a significant predictor of PPD. According to Fiala et al., there is a higher affinity for prenatal depression in pregnant women with lower education, but secondary education is considered to be a possible protective factor for developing PPD [31]. In a Vietnamese study, Do, Nguyen and Pham identified a lower level of education as a factor increasing the risk of PPD [64]. Since our study was not longitudinal, but a cross-sectional correlation study, we cannot speak of causality, but only of associations between variables. We recommend the implementation of other studies on this topic, especially longitudinal studies in post-communist countries, where this issue is explored to a lower rate.

## 5. Conclusions

The current study confirmed the association between the level of birth satisfaction and the risk of developing PPD, i.e., a lower level of satisfaction with childbirth was a significant predictor of a higher risk of PPD. Other risk factors that increased the risk of PPD were: stress (moderate stress), type of childbirth (caesarean delivery), parity (primiparity) and a positive psychiatric history. The same risk factors that were associated with the risk of developing PPD also negatively affected birth satisfaction. Further research on PPD predictors is needed, whose impact is contradictory, since some studies confirm the impact of PPD predictors, but others do not. These predictors include parity, i.e., primiparas vs. multiparas, and type of birth, i.e., vaginal vs. surgical delivery. In the case of surgical births, the type of birth should be consistently differentiated in the future, as, e.g., elective cesarean section may be more acceptable for a woman than emergency cesarean section. Support and care for women throughout labor and childbirth should consider the constant strengthening of the mother’s positive birth experience, which can act as a protective factor for the development of various postpartum complications, including the risk of developing PPD.

## Figures and Tables

**Table 1 ijerph-18-10458-t001:** Characteristics of participants and mean EPDS and BSS scores in different groups (Parity, Type of birth, Psychiatric history).

Total Number of Participants in the Study *n* = 584							
Variable		*n*	Mean (SD) Range %	EPDS *n* Score SD	*p*-Level	BSS *n* Score SD	*p*-Level
Age		583	30.6 (±4.9) 16–44				
Missing data		1					
University hospital birth center	Martin	381	65.2				
	Bratislava	203	34.8				
Education	Primary	13	2.23				
	Secondary	231	39.5				
	Higher	339	58.15				
Missing data		1					
Parity	Primipara	346	59.9	334 5.57 (±4.64)	0.020	316 109.87 (±15.36)	0.017
	Multipara	232	40.0	229 4.70 (±3.85)		191 113.24 (±15.42)	
Missing data		6					
Type of birth	Vaginal	431	73.8	422 4.80 (±4.24)	0.000	385 115.01 (±13.66)	0.000
	Operative	153	26.2	153 6.46 (±4.44)		127 99.41 (±14.70)	
Psychiatric history	Negative	556	95.4	550 5.06 (±4.19)	0.000	485 111.65 (±15.36)	0.003
	Positive	27	4.6	27 8.78 (±6.01)		26 102.58 (±14.33)	
Missing data		1					
Marital status	Married	366	62.7				
	Partner	201	34.4				
	Widowed	12	2.0				
	No partner	5	0.9				
Chronic disease in pregnancy	Yes	53	9.12				
	No	528	90.88				
Missing data		3					
Newborn disease	Yes	7	1.21				
	No	573	98.79				
Missing data		4					
Support person during birth	Partner	376	64.38				
	Significant other	33	5.65				
	None	175	29.27				
Preterm labor	Preterm	122	20.93				
	In term	259	44.43				
	Post term	202	34.64				
Missing data		1					
Gestational diabetes	Yes	29	4.97				
	No	555	95.03				
Abortion/Perinatal loss	Yes	128	77.97				
	No	453	22.03				
Missing data		3					
Financial status	Very good	251	43.20				
	Quite good	307	52.84				
	Not very good	23	3.96				
Missing data		3					
Religiosity participation	No	327	56.09				
	Sometimes	157	26.93				
	Regularly	99	16.98				
Missing data		1					

BSS (Birth Satisfaction Scale); EPDS (Edinburgh Postnatal Depression Scale).

**Table 2 ijerph-18-10458-t002:** Internal consistency of the BSS and subscales.

BSS and Subscales	Cronbach’s Alpha	*n* of Items
BSS	0.89	30
Subscales		
Quality of care	0.64	8
Personal attitudes	0.73	8
Stress experienced	0.84	14

**Table 3 ijerph-18-10458-t003:** Basic characteristics of BSS, EPDS and PSS.

Values	BSS	BSS Subscale 1	BSS Subscale 2	BSS Subscale 3	PSS	EPDS		EPDS Score > 10	EPDS Score > 13
	Total Score	Quality of Care	Personal Attributes	Stress Exper.	Total Score	Total Score		*n* %	*n* %
Mean	111.14	30.82	29.44	50.83	14.18	5.24	No	481 (83.36)	542 (93.93)
SD	15.45	4.07	5.01	8.51	5.68	4.35	Yes	96 (16.64)	35 (6.07)
Min.	61.00	18.00	15.00	24.00	0.00	0.00		-	-
Max.	149.00	40.00	40.00	69.00	30.00	26.00		-	-
*n*	512	532	561	562	566	577		577	577
Missing data	72	52	23	22	18	7		7	7

BSS (Birth Satisfaction Scale); EPDS (Edinburgh Postnatal Depression Scale); PSS (Perceived Stress Scale).

**Table 4 ijerph-18-10458-t004:** Linear regression analysis of postpartum depression risk factors in a research sample.

Model	Variable	Standardized β	95% CI	t	Sig.
**Model 1**	**Age**	−0.10	−0.18; −0.01	−2.10	0.036
	Education	0.05	−0.34; 1.25	1.12	0.264
	**Financial status**	0.14	0.37; 1.80	3.00	0.003
	Religiosity	0.05	−0.25; 0.79	1.02	0.307
	Marital status	0.09	−0.13; 4.42	1.85	0.065
	adjusted R^2^	0.03			
**Model 2**	Age	−0.03	−0.11; 0.05	−0.73	0.466
	Education	0.03	−0.46; 0.92	0.65	0.515
	Financial status	0.01	−0.51; 0.74	0.36	0.719
	Religiosity	0.04	−0.20; 0.71	1.09	0.277
	Marital status	0.05	−0.69; 3.20	1.27	0.206
	**Stress**	0.48	0.30; 0.43	11.44	0.000
	**Psychiatric history**	0.09	0.23; 3.34	2.26	0.025
	Chronic disease in pregnancy	0.00	−1.13; 1.25	0.10	0.919
	**Newborn disease**	0.09	0.56; 7.14	2.30	0.022
	Support person during birth	0.01	−0.71; 0.91	0.25	0.805
	Preterm labor	−0.02	−1.07; 0.67	−0.45	0.656
	**Parity**	−0.13	−1.98; −0.47	−3.20	0.001
	**Type of birth**	0.18	1.06; 2.64	4.60	0.000
	Gestational diabetes	0.05	−0.48; 2.67	1.37	0.172
	Perinatal loss	0.01	−0.76; 0.94	0.21	0.835
	adjusted R^2^	0.30			
**Model 3**	**BSS**	−0.18	−0.08; −0.03	−3.86	0.000
	**Stress**	0.43	0.27; 0.39	10.05	0.000
	**Psychiatric history**	0.08	0.01; 3.09	1.98	0.049
	Chronic disease in pregnancy	0.01	−0.98; 1.37	0.33	0.742
	Newborn disease	0.07	−0.33; 6.22	1.77	0.078
	Support person during birth	0.02	−0.59; 1.01	0.52	0.601
	Preterm labor	−0.03	−1.18; 0.55	−0.72	0.474
	**Parity**	−0.12	−1.82; −0.32	−2.81	0.005
	**Type of birth**	0.11	0.20; 1.94	2.41	0.016
	Gestational diabetes	0.06	−0.40; 2.70	1.46	0.144
	Perinatal loss	−0.00	−0.85; 0.83	−0.03	0.973
	Age	−0.02	−0.10; 0.06	−0.57	0.571
	Education	0.02	−0.48; 0.88	0.58	0.565
	Financial status	0.00	−0.62; 0.63	0.02	0.988
	Religiosity participation	0.05	−0.16; 0.74	1.28	0.203
	Marital status	0.05	−0.60; 3.24	1.35	0.176
	adjusted R^2^	0.32			

Bold in the table means the association between sociodemographic and clinical factors and the risk of postpartum depression.

## Data Availability

The data presented in this study are available on request from the corresponding author. The data are not publicly available due to ethical and privacy restrictions.

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
