# Peer review of "The Association between Birth Satisfaction and the Risk of Postpartum Depression"

_ijerph, 2021, doi:10.3390/ijerph181910458_

Round 1

Reviewer 1 Report

In general, the paper is organized and clear. However, I find that there are two main limitations:

  • There is some inconsistency throughout the paper regarding the focus of the study. The title and the Introduction are focused on the relationship between birth satisfaction and postpartum depression. However, the authors considered more variables which were not adequately contextualized in the Introduction. For instance, the rationale for selecting the variables mentioned in lines 84-86 should be included in the text. Furthermore, all statistical analyses should be consistent with the study aims. For instance, it was not an aim of the study to compare depression symptomatology and birth satisfaction in groups based on parity, type of birth and psychiatric history (presented in Table 1).
  • The authors did not demonstrate the relevance of this study. Considering that the relationship between the study variables was previously investigated and the results are mainly consistent across studies (as stated in lines 72-73), it is not clear what the paper adds to the current literature.

There are also some minor aspects that I suggest are revised by the authors:

  • The Introduction should be revised regarding the order of presentation of contents. I suggest that the authors start by defining birth satisfaction (e.g., currently in lines 62-68); than its antecedents (e.g., currently in lines 33-43; 56-59 – redundancies should be eliminated) and implications (e.g., currently in lines 43-44, 48-56), with a special emphasis on postpartum depression. Finally, the authors should present the study aims, highlighting its relevance for the state of the art.
  • The information on the “Setting and participants” subsection should be presented in chronological order (e.g., the authors should describe the sampling methods prior to describing the participants).
  • Redundant information should be deleted. Often, the authors presented the same data twice, in the text and in tables (e.g., lines 96-98; 160-177).
  • The authors should state whether permission was obtained from the Ethics Comittees of the hospitals where the data were collected.
  • The Birth Satisfaction Scale subscales were used in some analyses; as such, the internal consistency of each subscale in the current subscale should be presented.
  • The authors’ conclusions should be consistent with the statistical analyses. It is incorrect to state a causal relationship based on correlations (e.g., lines 198-200).
  • In line 206, the term “mediator” is incorrectly used. Statistically, this term is used to identify variables that explain the relation between two other variables; that is, mediators are mechanisms or processes.
  • Regarding the regression models, data regarding t values and significance values for each predictor should be presented. Also, the authors should explain how the dichotomous variables (e.g., parity, type of birth) were coded.
  • In the Discussion section, the authors should not repeat the results. The results should be interpreted, considering the results of previous studies and/or theories; also, study limitations, suggestions for future studies and clinical implications should be presented.
  • The verb “prove” should not used regarding results, given that scientific knowledge is tentative.
  • Considering that the authors did multivariate analyses, it is not useful to consider the relationship between a given predictor and the outcome in isolation or based on correlations.
  • In line 349, please revise this statement, as it is not clear.
  • Regarding the study limitations, the authors should reflect on the study design, considering the main aim of the study.
  • The authors should revise the text, as there are several errors regarding spacing (e.g., “in adequate” instead of “inadequate”; “recommendedfor” instead of “recommended for”).

Reviewer 2 Report

  1. The strenghts:
    1. The postpartum depression rate is underestimated and we, as a professional caregivers need a constant knowledge refreshing in this area.
    2. The survey was conducted among women giving birth in the country with communist history. Communism had a high, negative impact to maternity care. Postcommunist countries struggle with the quality of maternity care. It is very important to highlight this issue.
    3. The Birth Believes Satisfaction Scale is an important tool among women because of the rate of complicated pregnancies, pregnancy loss and cesarean sections. α Cronbach’s=0,79, for BBSMed is 0,89, for BBSNat is 0,94.
    4. EPDS is a reliable tool, widely used.
    5. PSS is a very good tool, using it in this case highlight the problem that the childbirth may be stressful for mothers.

All above mentioned scals were used properly and the relationship between the results was clearly  described.

  1. Weakness – it was a pleasure to read this article. I really can not indicate the significant weak points. Minor mistakes – some literal errors and editors mistakes in the lines: 56, 281, 299, 301.

Round 2

Reviewer 1 Report

I think the authors took some of my previous comments into consideration.

However, I believe that the paper should be improved regarding consistency. As the literature review in the Introduction is mainly concerned with the association between birth satisfaction and postpartum depression, I suggest that the authors consider only the aim of exploring this relationship, while controlling for relevant covariates (e.g., stress, clinical variables).
Considering this, I suggest that the authors eliminate section 3.3.

Additionally, regarding the regression models, two things should be considered:
- as some variables (e.g., stress, clinical variables) need to be controlled for in order to assess the influence of birth satisfaction on depression symptoms, such variables/covariates (not mediating variables, as stated in line 232) should be included in the first blocks - I suggest that demographic variables are included in the first block and clinical variables in the second; birth satisfaction should be added by itself in the last block, in order to assess its unique impact.
- considering that depression symptoms were assessed with the EPDS, which has a cut-off score, the authors should consider the possibility of using depression as a categorial variable (scores lower vs. higher than the cut-off score). If the authors choose to do so, a binary logistic regression model should be used. Also, the Discussion section should be revised accordingly.

I suggest some other changes to the manuscript:
- there is some redundancy in the information presented in lines 24-34; the authors should integrate this information in a single paragraph.
- the information in lines 35-38 concerns the impact of birth satisfaction; lines 39-48 are about the antecedents of birth satisfaction; lines 49-50 are about the consequences; lines 50-55 are about the antecedents; lines 56-65 are about the consequences. This informations needs to be better organized.
- lines 113: "paper-and-pencial questionnaire" should not be mentioned as an inclusion criteria; this was a data collection procedure.
- the authors should draw conclusions that are in accordance with the data. This was not always the case. For instance, in lines 440-441, the authors stated that "the same risk factors that were associated with the risk of developing PPD are also predictors of lower birth satisfaction". However, no regression models were used for predicting birth satisfaction.

Author Response

I think the authors took some of my previous comments into consideration.

Comment :

However, I believe that the paper should be improved regarding consistency. As the literature review in the Introduction is mainly concerned with the association between birth satisfaction and postpartum depression, I suggest that the authors consider only the aim of exploring this relationship, while controlling for relevant covariates (e.g., stress, clinical variables).
Considering this, I suggest that the authors eliminate section 3.3.

Author’s response:

Since we consider findings that we have reached in our research and presented in section 3.3 as important, we would like to keep the original version of the manuscript in section 3.3. The aim of the study has been modified with regard to other proposed changes.

Comment : Additionally, regarding the regression models, two things should be considered:
- as some variables (e.g., stress, clinical variables) need to be controlled for in order to assess the influence of birth satisfaction on depression symptoms, such variables/covariates (not mediating variables, as stated in line 232) should be included in the first blocks - I suggest that demographic variables are included in the first block and clinical variables in the second; birth satisfaction should be added by itself in the last block, in order to assess its unique impact.

Author’s response:

The regression model was modified according to the reviewer's recommendation: section 3.2. (relevant text and Table 4). At the same time, the Abstract and Data analysis 2.5. were changed.

I suggest some other changes to the manuscript:

Comment :

- there is some redundancy in the information presented in lines 24-34; the authors should integrate this information in a single paragraph.

Author’s response:

Redundant information has been removed (lines 32, 33).

Comment :

- the information in lines 35-38 concerns the impact of birth satisfaction; lines 39-48 are about the antecedents of birth satisfaction; lines 49-50 are about the consequences; lines 50-55 are about the antecedents; lines 56-65 are about the consequences. This informations needs to be better organized.

Author’s response:

We tried to organize the information in the Introduction better (lines 35 – 58).

Comment :

- lines 113: "paper-and-pencial questionnaire" should not be mentioned as an inclusion criteria; this was a data collection procedure.

Author’s response:

The words "paper-and-pencial questionnaire" has been removed (section 2.2. Setting and participants, lines 113 -- 114).

Comment :

- the authors should draw conclusions that are in accordance with the data. This was not always the case. For instance, in lines 440-441, the authors stated that "the same risk factors that were associated with the risk of developing PPD are also predictors of lower birth satisfaction". However, no regression models were used for predicting birth satisfaction.

Author’s response:

Thank you for your comment, the Conclusion of the manuscript has been modified in accordance with the data.